# Climate Change and Human Health: A Review of Allergies, Autoimmunity and the Microbiome

**DOI:** 10.3390/ijerph17134814

**Published:** 2020-07-04

**Authors:** Carly Ray, Xue Ming

**Affiliations:** Department of Neurology, Rutgers New Jersey Medical School, Newark, NJ 07103, USA; cnr28@njms.rutgers.edu

**Keywords:** climate change, allergy, autoimmunity, microbiome, dysbiosis, neurodevelopment

## Abstract

The impact of climate change on human health is a topic of critical importance. While only recently beginning to gain attention, it is clear that immediate action is necessary to minimize this impact. In our review, we will outline a subset of these effects in detail. We will examine how climate change has worsened respiratory allergic disease. We will discuss how climate change has altered antigen exposure, possibly disrupting antigen-specific tolerance by the immune system, leading, in turn, to an increase in the prevalence of immunologic diseases. Finally, we will explore how the loss of biodiversity related to climate change may affect the microbiome, potentially leading to dysbiosis, inflammatory, autoimmune and neurologic diseases.

## 1. Introduction

A warming climate has the potential to profoundly impact human health. The effects are widespread. As stated by the World Health Organization (WHO) Director-General in 2008, “climate change will affect, in profoundly adverse ways, some of the most fundamental determinants of health: food, air, water. In the face of this challenge, we need champions throughout the world who will work to put protecting human health at the centre of the climate change agenda” [1]. In this review, we will discuss in detail the relationship between climate change and allergies. While this topic has been previously discussed in the literature, we propose that there may be a more profound link between the two, i.e., that our changing climate may not only be worsening the condition of those with existing allergies, but also triggering allergies in those who are disease-naive. In addition, we will explore a possible link between current human practices and a declining biodiversity with disturbance to the human microbiome and increasing autoimmunity.

## 2. Allergy and Climate Change: What We Already Know

It is well documented that climate change leads to worsening symptoms in patients with established allergic disease. Patients with chronic respiratory allergic disease such as asthma and allergic rhinoconjunctivitis are at particular risk due to increased exposure to pollen, as well as the increased concentration and distribution of air pollutants [2].

As reported by the American Academy of Allergy Asthma & Immunology (AAAAI), climate change has both increased the intensity of the pollen season as well as prolonged its duration [3]. A retrospective study on allergenic pollen abundance and seasonality was published in the 2019 issue of The Lancet Planetary Health. Researchers analyzed global datasets containing data on pollen season duration and intensity for 17 locations across multiple continents in the northern hemisphere. The study found an increase in annual pollen load for 12 of the locations, as well as a significant increase in pollen season duration for 11 out of the 17 locations. Annual increases in maximum temperatures were significantly associated with an increase in pollen load, demonstrating a link between the two phenomena [4].

The heightened intensity of the pollen season can be attributed to the increased concentration of pollen. Climate change is, in part, a result of the increase in atmospheric greenhouse gases such as carbon dioxide, nitrous oxide and methane. In a review by Lewis et al. on atmospheric carbon dioxide and plant biology, increases in carbon dioxide were shown to lead to a reflexive increase in plant reproduction and total pollen levels. Carbon dioxide is one of four major resources required by plants for successful photosynthesis. As CO_2_ concentrations have risen, so too has the growth of plant species that thrive at high CO_2_ concentrations (“C_3_” plants) as well as their pollen byproducts [5]. Ragweed pollen for example, one of the major causes of respiratory allergy, has been increasing in concentration, with models predicting that levels will increase by 4 times compared to the current values within the next 30 years [6]. Thunderstorms, which have become more frequent due to rising sea temperatures, have been found to increase concentrations of pollen grains at ground level. After absorbing water, these grains can rupture from osmotic shock, releasing allergenic particles that can induce severe asthmatic symptoms in patients with asthma, hay fever and allergic rhinitis [7].

As described by Katelaris et al., clinical evidence showed that the aforementioned changes in pollen led to increased respiratory allergic disease, evident from correlations between pollen and respiratory disease exacerbations [2]. A recent study performed in Melbourne, Australia found a dose-response association between ambient grass pollen levels and the same day risk of childhood asthma presentations to the emergency department [8].

Climate change has also been linked to increased concentrations and distribution of air pollutants such as ozone, nitric oxide and other volatile organic chemicals. There is a growing body of evidence suggesting that these airborne environmental pollutants may be partially responsible for the substantial increase in allergic respiratory disease seen in industrialized countries over the past several decades [9]. The results of several studies suggest that the inhalation of air pollutants may potentiate the airway response to inhaled allergens in persons predisposed to atopic disease. The inhalation of air pollutants, notably diesel exhaust particles (DEPs), sulfur dioxide and nitrogen dioxide, has been found to cause inflammation and increased mucosal permeability within the airways, ultimately allowing increased barrier penetrance by allergens. Oxidative stress is thought to be the major mechanism through which this occurs. Oxidants produced by pollutants within respiratory mucosal epithelial cells trigger the release of inflammatory mediators, causing the destruction and apoptosis of respiratory mucosal cells, and eventually leading to bronchial hyperreactivity. In addition to increasing mucosal permeability, DEPs in particular can bind to pollens, animal and other allergens, directly facilitating their entry into the respiratory epithelium [10].

Early exposure to air pollutants has also been linked to an increased risk of incident asthma, allergic rhinitis and eczema in children. A follow-up to the Toronto Child Health Evaluation Questionnaire (T-CHQ) study found that exposure to ozone and nitrogen dioxide at birth was associated with a 17% increased risk for developing asthma at around 4 years of age, and a 7% increased risk for developing eczema at 3.5 years [11]. A systematic review and meta-analyses of birth cohort studies examining the impacts of childhood exposure to traffic-related air pollution (TRAP) found that early exposure to particulate matter was associated with increased risk of asthma, sensitization to aero and food allergens, and the development of eczema and hay fever [12].

What we know: We know that climate change has worsened allergies in those with established respiratory allergic disease.

## 3. Immunologic Disease and Climate Change: Is There a Deeper Link?

As described above, there is ample evidence supporting the link between climate change and worsening respiratory allergic disease. We hypothesize that the relationship between climate change and immunologic diseases such as allergies may be more extensive, i.e., that a warming climate may result in an erratic immune system and consequently, the emergence of allergic and autoimmune disease in the disease-naïve population.

The immune system has the ability to differentiate pathogenic antigens from benign antigens found in our everyday environment. When an antigen is identified as innocuous, the body has the ability to suppress the immune system from initiating an inflammatory response. Our understanding of this mechanism of tolerance remains incomplete; however, various models have been proposed. As explained by Vickery et al., the gastrointestinal tract is continuously exposed to an extremely diverse profile of antigens. These include benign environmental antigens such as food proteins, harmful exogenous pathogens and commensal microbiota. The gut has a system for the antigen-specific suppression of immune response that involves T-cells, antigen presenting cells, cytokines, B-cells and antibodies [13]. The development and maintenance of this immune tolerance is critical, as loss or failure of tolerance can result in disorders such as autoimmunity, allergy, respiratory illnesses such as rhinitis or asthma, and even irritable bowel syndrome. As described in our prior article [14], climate change has dramatically altered the profile of antigens our bodies are exposed to. While many of these antigens are benign, it is possible that the extent of this change has overwhelmed the immune system’s long-standing ability for antigen-specific tolerance, and the increasing molecular mimicry due in part to increasing exposure to allergens, is leading to a rise in immunologic disease.

Analysis of CDC surveillance and reporting data shows that immunologic disorders are on the rise [15,16]. Food allergies are one example. Recent data from Northwestern University showed that 10.8% of adults in the US have a food allergy, with 48% of those developing at least one such allergy as an adult [17]. According to research from the American College of Allergy, Asthma and Immunology, peanut allergy in children increased by 21 percent from 2010 to 2017 [18]. In a 2016 New Jersey Monthly article, Joanna Buffum outlined the ways in which the increase in food allergies among children has changed the public school experience. There are now rules regulating food, including making classrooms peanut-free. In 2013, the CDC published the first comprehensive guidelines for the management of food allergies in schools. As stated by one mother, “kids didn’t have these problems when I was young”. Studies investigating the cause for this surge are underway [19]. The notion that climate change may be one of the inciting factors is supported by several studies which have found that increases in CO_2_ and temperature are correlated with changes in the composition of peanut (*Arachis hypogaea*), potentially altering its antigenicity [20].

The deeper link: Climate change has altered our antigen exposure, potentially disrupting antigen-specific tolerance by the immune system and increasing the incidence of molecular mimicry within the body. This may explain in part the surge in immunologic disease that has been observed in recent years.

## 4. A Shifting Environment and the Microbiome: Dysbiosis and Inflammatory Diseases

The human gastrointestinal tract consists of an estimated 100 trillion microbes from at least 160 different species. Studies have shown that the composition of the microbiota fluctuates early in life, influenced by external factors such as one’s environment and diet. As an individual reaches adulthood, their microbiome profile typically stabilizes. It has become clear that these commensal bacteria play an active role in human immunity, from the maintenance of barrier defense to the development of immune tolerance. There is a growing body of literature demonstrating that disruption of the microbiome, known as dysbiosis, can lead to a variety of secondary effects within the body. Gut microbiome dysbiosis can lead to the development of infectious, inflammatory and autoimmune disease. Loss of biodiversity as a result of climate change will be explored as a possible vector of this dysbiosis.

### 4.1. The Effects of Dysbiosis

The microbiota plays an important role in the maturation of the immune system. Through processes such as toll-like receptor recognition of microbe-associated molecular patterns (MAMPs), the body learns to differentiate between commensal microbes and pathogenic bacteria. In addition, gut microbes are involved in T-cell differentiation, T-regulatory and T-helper 2 responses, and the maintenance of gut epithelial integrity [21,22]. Perturbations of intestinal microbiota, i.e., dysbiosis, can lead to the dysfunction of these processes and increase the risk for multiple inflammatory and autoimmune disorders. This dysbiosis-induced immune dysregulation is supported by studies showing that patients with these diseases have microbiota profiles that differ from healthy controls. For example, one study found a four-fold increase in the concentration of *Ruminococcus gnavus* in the intestinal epithelia of patients with both Crohn’s disease and Ulcerative Colitis, compared to healthy controls. *R. gnavus* has been linked to other inflammatory diseases including spondyloarthritis and infantile eczema [23,24].

The relationship between dysbiosis and allergy has been well established. Observational studies have demonstrated that disruption of gut colonization early in life, for example due to cesarean section or lack of breastfeeding, can lead to altered microbiota profiles and increased incidence of allergic disease [25]. Reports have also implicated dysbiosis in the development of food allergy, with several studies finding differences in the microbial makeup of children with food allergies compared to healthy controls [26]. A cross-sectional study found increased levels of *Clostridium* sensu stricto and *Anaerobacter*, and decreased levels of *Bacteroides* and *Clostridium XVIII* in children with food allergies [27]. A recent prospective study found that low gut microbial richness at 3 months of age preceded food sensitization at 12 months [28].

The effect of dysbiosis on the human body is not limited to the development of autoimmune and inflammatory diseases. Murine models have implicated the microbiota in CNS development, and dysbiosis has been linked to neurologic diseases such as multiple sclerosis, autism and Parkinson’s disease (PD). Patients with PD have been found to have abnormal microbiota profiles, and current research suggests that gastrointestinal dysfunction may be a trigger for alpha-synuclein aggregation in neurons [29]. Microbiome imbalance and increased gut permeability have been implicated in autism spectrum disorder (ASD). In my own research, I found abnormal amino acid metabolism, increased oxidative stress, and altered gut microbiomes among a subset of patents with ASD [30]. It has been suggested that the so-called gut–brain axis may play a role in several neurodevelopmental disorders.

### 4.2. Microbiome and Biodiversity Loss

A 2019 global report released by the Intergovernmental Science-Policy Platform on Biodiversity and Ecosystem Services (IPBES) announced that biodiversity “is declining faster than at any time in human history”, claiming that nearly 1 million animal and plant species are threatened with extinction. Climate change was cited as a leading factor for this decline, with changes in land and sea use and pollution also being implicated [31]. With this decline in macrodiversity comes an accompanying decline in microdiversity [32].

The “biodiversity hypothesis” suggests that reduced contact between people and a biologically diverse natural environment may adversely affect the microbiota and its immunomodulatory capacity [32]. Many members of the scientific community support this hypothesis, such as The World Allergy Organization (WAO), who declared that “biodiversity loss leads to reduced interaction between environmental and human microbiotas, which in turn may lead to immune dysfunction and impaired tolerance mechanisms.” In their statement, the WAOJ suggested that these changes may account in part for the increase in the prevalence of asthma, allergy and inflammatory disease in the developed world [33]. Von Hertzen et al. outlined a mechanism for this hypothesis, suggesting that microbial deprivation can lead to impaired immunoregulatory circuits. In the absence of adequate microbial stimuli through the skin, gut and respiratory tract, IL-10, TGFβ, regulatory dendritic cell and regulatory T-cells (T_REG_) are not sufficiently induced, leading to an inflammatory environment. In such an environment, T_REG_ cells are converted to T-helper 17 cells (T_H_17), ultimately enriching bacteria that tolerate inflammatory mediators [34].

Studies dating back to 2005 support this hypothesis, showing that exposure to rich microbial environments correlates with protection against the development of future allergic and autoimmune disease [35,36]. When compared to a reference group, school aged children living on farms were found to be exposed to a wider range of microbes and have lower prevalences of asthma and atopy [37]. There is also evidence that, compared to healthy controls, patients with atopy tend to reside in areas of lower environmental biodiversity (i.e., plant species, land types). In one study, individuals with atopic disease were found to have reduced diversity of cutaneous gammaprotectobacteria, a bacteria found in the soil and above-ground vegetation. There was a significant correlation between amounts of gammaproteobacteria and IL-10 expression in healthy participants, suggesting that the bacteria may have a protective effect against allergy [32].

Climate, current practices and the gut: The microbiome is a key player in the development and maintenance of the immune system. It has also been implicated in central nervous system development. Dysfunction as a result of a lack of immersion in biodiverse natural surroundings can lead to serious inflammatory, autoimmune and neurologic diseases.

## 5. Future Steps

In the above review, we have demonstrated various ways in which climate change has the potential to affect the human body. To minimize this impact, we must end the destruction of our natural environment, decrease emissions of greenhouse gases and urge our peers to adopt more “green” behavior. We have also explored the impact of broader shifts in human practices such as decreased exposure to nature and the use of antimicrobials on subsequent health and disease development. With research demonstrating links between the microbiome and autoimmune, inflammatory, and neurologic diseases, it is critical that we minimize antimicrobial exposure. This may involve altering guidelines for the prescription of antibiotics by medical professionals. In addition, given that the microbiome is directly impacted by our daily environment, it is important to regularly immerse ourselves in nature and familiarize ourselves with biodiverse surroundings.

## 6. Conclusions

It is clear that the effects of climate change extend beyond the natural environment and into the human body. In this review, we have detailed several of these effects. A warming climate has caused exacerbation of existing respiratory allergic diseases and is implicated in the surge of allergies and autoimmunity in those who were previously disease naive. Climate change has dramatically altered the profile of antigens our bodies are exposed to, bombarding our immune systems and potentially overwhelming its ability for antigen-specific tolerance. More broadly, shifts in current human practices such as the use of antimicrobials and decreased immersion in biodiverse surroundings have effects on the microbiome. Dysbiosis has serious implications, as has been linked to the development of inflammatory, autoimmune and neurologic disease. Climate change is no longer an issue for future generations. It is affecting every one of us, now. If we do not act immediately, its impact on human health will undoubtedly worsen. Our bodies have the incredible ability to adapt to environmental change; however, adaptation takes generations. It is imperative that we work to halt climate change, not only to save our planet, but also to save our own lives.

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
