# Peer review of "Climate Change and Human Health: A Review of Allergies, Autoimmunity and the Microbiome"

_ijerph, 2020, doi:10.3390/ijerph17134814_

Round 1

Reviewer 1 Report

ijerph-803082-peer-review-v1

title: Climate Change and Human Health: A Review on  Allergy, Autoimmunity and the Microbiome

Dear Editor

The manuscript is interesting , but it needs some of the  following points to be clarified before being published.

Minor Criticisms:

  • Paragraph 2 entitled: Allergy and Climate Change: What we already know, line 58 ... substantial increase in allergic respiratory disease seen in industrialized countries over the past several decades [8]... PLEASE SPECIFY which is the  factors and link between urban pollution and allergy i.d.: increased concentrations of air pollutants and pollens can activate inflammatory mediators in the airways, etc. as reported in recent references (i.g. Clin Mol Allergy. 2018 Sep 11;16:20. doi: 10.1186/s12948-018-0098-3. eCollection 2018);
  • Paragraph 2 entitled: Allergy and Climate Change: What we already know, line 64... incident asthma and eczema in children.... PLEASE COMPLETE with ... incidence of childhood asthma, allergic rhinitis and eczema );
  • Paragraph 4.2 entitled: Material and Methods, after line 162 ... Studies dating back to 2005 have shown that exposure to rich microbial environments correlates with  protection against the development of future allergic and autoimmune disease... USEFULL TO ADD  a refence  or PLEASE CLARIFY.
  • It is relevant and interesting, Interesting aspects of knowledgement in this area
  • The topic is quite original.
  • The text is clear and easy to read.
  • The conclusions are consistent with the evidence and arguments presented

Author Response

The following is the point by point response to the reviewer's critiques. The authors are grateful to this reviewers critical insights!

Paragraph 2 entitled: Allergy and Climate Change: What we already know, line 58 ... substantial increase in allergic respiratory disease seen in industrialized countries over the past several decades [8]... PLEASE SPECIFY which is the  factors and link between urban pollution and allergy i.d.: increased concentrations of air pollutants and pollens can activate inflammatory mediators in the airways, etc. as reported in recent references (i.g. Clin Mol Allergy. 2018 Sep 11;16:20. doi: 10.1186/s12948-018-0098-3. eCollection 2018);

Response: To clarify the link between pollution and allergic response we included an additional paragraph and citation detailing mechanisms of inflammation and bronchial hyperreactivity.

Paragraph 2 entitled: Allergy and Climate Change: What we already know, line 64... incident asthma and eczema in children.... PLEASE COMPLETE with ... incidence of childhood asthma, allergic rhinitis and eczema );

Response: We have completed with “allergic rhinitis and eczema.”

Paragraph 4.2 entitled: Material and Methods, after line 162 ... Studies dating back to 2005 have shown that exposure to rich microbial environments correlates with protection against the development of future allergic and autoimmune disease... USEFULL TO ADD  a refence  or PLEASE CLARIFY.

Response: We have added the following references: “Vartiainen E, Petays T, Haahtela T, Jousilahti P, Pekkanen J. Allergic diseases, skin prick test responses, and IgE levels in North Karelia, Finland, and the Republic of Karelia, Russia. J Allergy Clin Immunol 2002;109:643-648”;  “von Mutius E, Radon K. Living on a farm: impact on asthma induction and clinical course. Immunol Allergy Clin North Am 2008;28:631-647; doi:10.1016/j.iac.2008.03.010.”

Reviewer 2 Report

Overview, while the sections present some interesting findings, more work is needed to expand the subsections and the scope of the review as well as to link the sections together. For example, the sudden mention of human practices and microbiome is quite abrupt and does not seems linked to the topic on climate change. Rather, evidence should be presented on how climate change affect microbiome dysbosis and subsequent health outcomes.

Under the section of allergy and climate change:

Lines 30-31  Reference needed for the sentence :” patients with chronic respiratory allergic disease are a demographic at particular risk of … “

Climate change can be attributed to a number of factors such as global warming, traffic related air pollution. Besides this, thunderstorm-related allergic respiratory diseases should be discussed too.

Mechanisms of how climate change affect human health should also be discussed in more details- for example does this affect inflammation? Oxidative stress? Or could it be influenced by the impact of air particulate matter on gut microbiome ? More findings from studies of other groups is needed.

A table or figure summarising this evidence will be helpful

  1. Immunologic Disease and Climate Change: Is There A Deeper Link?

The authors discussed about food allergy in particular , how about the respiratory allergic diseases? The link between climate change and subsequent immunologic disease development as well as mechanisms is also not explained in detail.

4. A Shifting Environment and The Microbiome: Dysbiosis and Autoimmunity

The section on the use of anti-microbials and microbiome dysbiosis is out of place in this review. Could it be that human practices result in climate change with subsequent impact on the host micobiome? This link should be established too. However, it seems that human practices such as deforestation and use of motor vehicles affects climate change rather than use of antimicrobials and antibiotics.

The section on biodiversity and microbiome dysbiosis is relevant but efforts should be placed on linking climate change to loss in biodiversity with subsequent impact on microbiome dysbiosis. Also it will be great to show results from animal models too, for example those who have inhaled particulate matter and the impact on their gut microbiome to show the direct effect of climate changes on gut microbiome. In addition, more human studies are needed to show this relationship, the possible mechanisms should be discussed too.

Author Response

The following is the point by point response to the reviewer's critiques. The authors are grateful to this reviewers critical insights!

Under the section of allergy and climate change:

Lines 30-31  Reference needed for the sentence :” patients with chronic respiratory allergic disease are a demographic at particular risk of … “

Response: It is now referenced [2]. 

Climate change can be attributed to a number of factors such as global warming, traffic related air pollution. Besides this, thunderstorm-related allergic respiratory diseases should be discussed too.

Mechanisms of how climate change affect human health should also be discussed in more details- for example does this affect inflammation? Oxidative stress? Or could it be influenced by the impact of air particulate matter on gut microbiome ? More findings from studies of other groups is needed.

A table or figure summarising this evidence will be helpful

Response: We have added two paragraphs to address the concerns. We do not make a table or figure as the information cited was descriptive, and the mechanisms of immune intolerance are by in large hypothetical. 

  1. Immunologic Disease and Climate Change: Is There A Deeper Link?

The authors discussed about food allergy in particular , how about the respiratory allergic diseases? The link between climate change and subsequent immunologic disease development as well as mechanisms is also not explained in detail.

Response: very good point indeed. Respiratory allergic diseases and their similar mechanisms are now inserted

4. A Shifting Environment and The Microbiome: Dysbiosis and Autoimmunity

The section on the use of anti-microbials and microbiome dysbiosis is out of place in this review. Could it be that human practices result in climate change with subsequent impact on the host micobiome? This link should be established too. However, it seems that human practices such as deforestation and use of motor vehicles affects climate change rather than use of antimicrobials and antibiotics.

Response: Air pollutions from carbon overproductions on human health is emphasized. 

The section on biodiversity and microbiome dysbiosis is relevant but efforts should be placed on linking climate change to loss in biodiversity with subsequent impact on microbiome dysbiosis. Also it will be great to show results from animal models too, for example those who have inhaled particulate matter and the impact on their gut microbiome to show the direct effect of climate changes on gut microbiome. In addition, more human studies are needed to show this relationship, the possible mechanisms should be discussed too.

Response: this manuscript is now strengthened by additional discussion on biodiversity. 

Round 2

Reviewer 2 Report

I appreciate the revisions that the authors have made. After another review,  I have some minor comments detailed below as well as suggest the following major comments:

Major comment :

  • In 4 : A shifting environment and the microbiome : Dysbiosis and autoimmunity- should be changed to dysbiosis and inflammatory diseases since the focus on this subsection is on proinflammatory diseases instead of autoimmunity. In addition, more evidence should be presented on allergic disease, there is extensive literature in the field on gut microbiome dysbiosis between allergic and non-allergic children. It will be great to link up with the changes in allergenicity as mentioned in the earlier section too to complete the picture.
  • Even if hypothetical, it is also useful to have a figure to illustrate the possible effects of climate change and human practices on microbiome dysbiosis and human. 

Minor comments

Line 32 Grammar structure : climate change leads to…

Line 34 : Grammar structure : allergic rhinoconjuntivitis are at particular risk

Lines 47: The title of the article should be removed and sentence changed to “ in a review by Lewis et al on atmospheric carbon dioxide and plant biology, rise in carbon dioxide led to…

Lines 59 : The article name should be removed and sentence changed to clinical evidence showed that aforementioned changes in pollen led to increased respiratory allergic disease, evident from correlatione between pollen count with respiratory  disease exacerbations

Lines 83 : What timepoints are asthma and eczema evaluated at?

Lines 101 : reference is missing

Line 103 : antigen presenting cells

Lines 107-108 : please remove article title

Line 119: please remove article title

Line 137 : reaches adulthood,

Line 140 : disruption of microbiome, known as dysbiosis,

Line 141 : Gut microbiome dysbiosis can lead to dysfunction …

Line 152: remove mechanism- this dysbiosis induced immune dysregulation

Line 156 : Rumincoccus gnavus should be in italics, in the intestinal epithelium

Line 156 : R.gnavus in italics

Line 181- WAoJ suggested

Line 195 : reduced diversity instead of less diverse array

Line 198 : protective effect against allergy

Line 207 : affect instead of effect

Line 210 : use of antimicrobials on subsequent health and disease development.

Line 213 : Not sure what does it mean by use of more commonplace chemical detergents and antimicrobials by the general public

Line 227 : affecting and not effecting. If we do not act immediately, ..

Line 229 change,
